# Molecular Aspects and Prognostic Significance of Microcalcifications in Human Pathology: A Narrative Review

**DOI:** 10.3390/ijms22010120

**Published:** 2020-12-24

**Authors:** Rita Bonfiglio, Annarita Granaglia, Raffaella Giocondo, Manuel Scimeca, Elena Bonanno

**Affiliations:** 1Department of Experimental Medicine, University of Rome “Tor Vergata”, via Montpellier 1, 00133 Rome, Italy; rita.bonfiglio@uniroma2.it (R.B.); raffaellagiocondo@gmail.com (R.G.); 2Fondazione Umberto Veronesi (FUV), Piazza Velasca 5, 20122 Milano, Italy; 3‘Diagnostica Medica’ & ‘Villa dei Platani’, Neuromed Group, 83100 Avellino, Italy; a.granaglia@gmail.com; 4Department of Biomedicine and Prevention, University of Rome “Tor Vergata”, via Montpellier 1, 00133 Rome, Italy; manuel.scimeca@uniroma2.it; 5San Raffaele University, via di Val Cannuta 247, 00166 Rome, Italy; 6Saint Camillus International University of Health Sciences, via di Sant’Alessandro, 8, 00131 Rome, Italy

**Keywords:** microcalcifications, ectopic calcifications, EMT, osteoblast-like cells, breast cancer, thyroid, lung, ovary

## Abstract

The presence of calcium deposits in human lesions is largely used as imaging biomarkers of human diseases such as breast cancer. Indeed, the presence of micro- or macrocalcifications is frequently associated with the development of both benign and malignant lesions. Nevertheless, the molecular mechanisms involved in the formation of these calcium deposits, as well as the prognostic significance of their presence in human tissues, have not been completely elucidated. Therefore, a better characterization of the biological process related to the formation of calcifications in different tissues and organs, as well as the understanding of the prognostic significance of the presence of these calcium deposits into human tissues could significantly improve the management of patients characterized by microcalcifications associated lesions. Starting from these considerations, this narrative review highlights the most recent histopathological and molecular data concerning the formation of calcifications in breast, thyroid, lung, and ovarian diseases. Evidence reported here could deeply change the current point of view concerning the role of ectopic calcifications in the progression of human diseases and also in the patients’ management. In fact, the presence of calcifications can suggest an unfavorable prognosis due to dysregulation of normal tissues homeostasis.

## 1. Introduction

The presence of calcium deposits in human lesions is largely used as imaging biomarkers of human diseases such as breast cancer [1,2,3]. Indeed, the presence of micro- or macrocalcifications is frequently associated with the development of both benign and malignant lesions. Several scoring systems are used in clinical practice in order to assess the risk of the presence of a malignant lesion associated with the imaging features of calcifications. In this context, the most popular clinical scoring systems have been developed for evaluating coronary and breast calcifications [4,5]. Despite the essential clinical role of microcalcifications in the assessment of breast cancer and cardiovascular diseases, the cellular and molecular mechanisms involved in their formation are still not completely known. Only in recent years have some aspects of microcalcification formations been elucidated. 

These investigations reported possible molecular mechanisms involved in the deposition of calcium crystals, mainly the epithelial to mesenchymal transition (EMT) phenomenon and osteoblastic differentiation [6,7,8,9,10]. Moreover, authors demonstrated that calcium deposits detected in human tissues can be produced by different mechanisms both active and associated to degenerative processes [11,12,13]. This evidence is also proven by the presence of different calcifications in terms of both histological appearance and elemental composition. Thus, it was found that the term microcalcifications includes very different types of calcium deposits, such as calcium oxalate (CO) or hydroxyapatite (HA), that are linked to distinct molecular mechanisms and prognostic significance [14,15,16,17]. Further complicating the matter is the presence of microcalcifications in numerous biological contexts—such as normal tissues, cancer, and benign conditions—and also in different organs (e.g., breast, lung, thyroid, and ovary). 

Therefore, a better characterization of the process related to the formation of microcalcifications in different tissues and organs, as well as the understanding of the prognostic significance of the presence of these calcium deposits into human tissues could significantly improve the management of patients characterized by microcalcifications associated lesions. 

Starting from these considerations, this narrative review highlights the most recent histopathological and molecular data concerning the formation of calcifications in breast, thyroid, lung, and ovarian diseases.

### Molecular Mechanisms Involved in the Formation of Microcalcifications

Historically, microcalcifications were considered a sign of tissue degeneration frequently associated with cancer necrosis. Thus, few studies have been performed to elucidate the molecular mechanisms involved in the formation of these elemental lesions. Anyway, it has been known for a long time that microcalcifications exist in several forms, considering both their histological appearance and elemental composition [14,16,18]. As mentioned above, the histological analysis describes three different forms of microcalcifications: birefringent crystals, mainly composed of CO; psammoma bodies (PBs); and dystrophic calcifications generally composed of HA and/or magnesium-substituted hydroxyapatite (Mg-HAp) [15,19].

The presence of calcifications with different elemental composition suggests the existence of different mechanisms involved in their formation. Indeed, while the presence of CO can be associated to both the degenerative process of pathological tissues and the deregulation of cell calcium homeostasis, the formation of complex forms of calcifications, such as HA and/or Mg-HAp, is generally considered an active process carried out by real osteoblasts.

In this scenario, Morgan et al. performed pioneering studies in which some molecular mechanisms involved in the formation of breast microcalcifications have been elucidated [20,21]. Morgan and colleagues demonstrated that the formation of microcalcifications by breast cancer cell lines was similar to the mineralization process that occurs in bone [22]. Breast cancer cells stimulated with osteo-inductive factors acquired the capability to form HA crystals [23].

Starting from this evidence, our group investigated the molecular aspects of the formation of breast microcalcifications on human biopsies. Thanks to these investigations, we described for the first time the breast cells responsible for the formation of calcium crystals made of HA or Mg-HAp [24]. Morphological, ultrastructural, and in situ molecular analysis revealed a surprising similarity between breast cancer cells capable of producing calcium crystals and the real osteoblasts [25]. Specifically, during the formation of microcalcifications, breast cancer cells showed morphological characteristics of osteoblasts such as large endoplasmic reticulum, large nucleus, and electron-dense granules. In addition, these cells were characterized by the expression of the main molecules involved in the calcium homeostasis such as receptor activator of nuclear factor kappa-B ligand (RANKL), runt-related transcription factor 2 (RUNX2), vitamin D receptor (VDR), and pentraxin-related protein 3 (PTX3) [26]. Due to the morphological and molecular similarity with the osteoblasts, we coined the term breast osteoblast-like cells (BOLCs).

Further investigations demonstrated the BOLCs originate from epithelial cells, undergoing the EMT phenomenon that, under bone morphogenetic protein-2 (BMP-2) and/or PTX3 stimulus, differentiate into osteoblast like cells. Thus, the production of microcalcifications by BOLCs can be considered a late sign of the cancerous transformation related to EMT [27]. In addition, it is known that EMT prepares the tumor cells for invading the surrounding tissues and forming distant metastasis. This evidence allows us to speculate that the presence of microcalcifications produced by BOLCs could represent a risk factor for both breast cancer occurrence and development [28].

Similar investigations have been performed about the molecular mechanisms involved in the formation of prostate calcifications [29]. In fact, the production of calcifications made of HA in prostate cancer seems to be related to the activity of epithelial cells with an osteoblast appearance named prostate osteoblast-like cells (POLCs). In addition, Scimeca et al. demonstrated that the presence of HA calcifications and POLCs in prostate cancer lesions significantly increase the risk of bone metastasis formation [30].

As for breast microcalcifications, for several years atheromatic calcifications have been studied considering only their prognostic value. Indeed, calcifications into the atheroma are commonly investigated by imaging analysis to establish the risk of plaque rupture and relative clinical events such as stroke [31]. More recently, several studies also investigated the mechanisms involved in the formation of these calcium deposits during the development of atheromatic plaques both in carotid and coronary arteries. Calcification has been determined to be an active process driven in part by vascular smooth muscle cell (VSMC) trans-differentiation within the vascular wall. VSMC can switch their morphological and immune characteristics, adopting a physiological contractile phenotype or an alternate ‘synthetic’ phenotype during the atheroma formation. More specifically, it has been demonstrated that VSMCs are able to adopt a number of phenotypes, including calcific (osteogenic, chondrocytic, and osteoclastic), adipogenic, and macrophagic phenotypes, suggesting a very predominant role of these cells in both the atheroma and calcification formation [32].

As concerns the calcifications, they can modify portions of the vessel conduit to a noncompliant structure. These calcium deposits may also alter tensile forces on the cap of a fibroatheroma, thus influencing the plaque stability [33]. Several local and systemic factors are involved in the formation of vessel calcifications, including hyperlipidemia, ongoing inflammation, large necrotic cores, and diabetes. Under the influence of osteo-inductive molecules—such as BMPs, PTX3, or Vitamin D—vascular cells and circulating hematopoietic stem cells undergo osteoblastic differentiation, inducing the deposition of calcium crystals [34]. In this context, matrix vesicles secreted by vessel osteoblast-like cells (VOLCs) in the extracellular space of the necrotic core serve as a nidus for calcification.

It is known that the size of calcifications correlates with the plaque stability mainly in the coronary lesions [35]. Indeed, even if coronary calcification can be considered a marker of coronary atheroma, dense calcifications (>400 HU) are generally associated with stable plaques. On the contrary, microcalcifications (often also referred to as spotty calcification) are more commonly found in coronary vulnerable plaques. Recently, it has been proposed that both the proximity of one microcalcific locus to another, and the orientation of the microcalcification about the blood flow could increase local tissue stress, destabilizing the atheromatic plaque [13].

Currently, four mechanisms have been described for early phases of the formation of vessel calcifications [36]: (a) aggregation of apoptotic bodies and necrotic debris released by inflammatory cells in the atheroma that provide a substrate for calcium phosphate crystal formation, (b) release of matrix vesicles by VOLCs or the accumulation of circulating nucleation complexes that serve as sites for calcium complex crystallization, (c) impairment of the expression of the main inhibitors of the mineralization process, and last but not least, (d) the appearance of vessel osteoblast-like cells from the differentiation of pericytes, VSMCs or circulating hematopoietic stem cells [37,38,39].

In carotid endarterectomy samples, matrix vesicles are frequently observed in atheromatic plaque acellular areas rich in elastic fibers [40,41,42]. These vesicles, as well as the apoptotic bodies released during the death of inflammatory cells or VSMCs, can contribute to the early stages of both intimal and medial calcification [43,44,45]. According to their size and type, these vesicles are classified as either ectosomes (microparticles 50 to 1000 nm; rich in cholesterol and diacylglycerol), exosomes (40 to 100 nm; rich in cholesterol, sphingomyelin, and ceramide), apoptotic bodies (50 to 5000 nm), or matrix vesicles (30 to 300 nm) [46]. In addition, it is known that extracellular vesicles include a large variety of cargoes as lipids, proteins, and micro–ribonucleic acid [40]. Vesicle nucleation sites allow deposition of calcium orthophosphate [47,48], which progresses to amorphous calcium phosphate [49,50] and then to more crystalline structures such as HA.

All of this evidence supports the idea that the formation of calcifications in human tissues is frequently an active and fine regulated process (Figure 1) which may reveal a dysregulation of tissue homeostasis.

## 2. Microcalcifications in Breast Cancer

Microcalcifications play a crucial role in early breast cancer diagnosis. In fact, approximately 50% of non-palpable breast cancers are detected by mammography only through microcalcifications patterns [27,51], but mammography cannot distinguish type I and II microcalcifications. However, some classification systems, such as Breast Imaging Report and Data System (BIRADS) [52], associate the shape, density, and distribution within the breast of the microcalcifications to assess the risk of the presence of a malignant lesion. Nevertheless, a tissue biopsy should be performed to confirm if the microcalcifications seen on mammography are associated with cancer. Analysis of microcalcifications by electron microscopy, microprobe analysis, and X-ray diffraction, has revealed two distinct forms of microcalcifications in breast disease based on their appearance and chemical composition [21].

Historically, the chemical composition of microcalcification is related to two main types of calcium deposits: CO, often associated with benign lesions, are amber-colored, partially transparent, and form pyramidal structures with relatively planar surfaces, and calcium phosphate—mainly HA—related both to benign and malignant lesions: they are grey-white, opaque with ovoid or fusiform shapes, and have irregular surfaces [19,53]. Under microscope, microcalcifications may be distinguished into three types: polymorphous bodies, birefringent crystals, and PBs. Several points of evidence suggest that the morphological appearance of microcalcifications in mammographic images—such as size, shape, and distribution—is associated with patient prognosis [24]. Many literature studies support the presence of a relationship between the histological appearance of the tumor (malignant or benign) and the chemical nature of the calcium deposit. Busing et al. reported that breast cancer lesions with microcalcifications composed of CO were found only in proliferating but non-invasive breast diseases, for example, proliferative mastopathy or lobular carcinoma in situ (LCIS), but HA crystals were found only in invasive malignant tumors like undifferentiated ductal carcinoma [21,54]. In this regard, it has been shown that HA can directly increase the malignant potential of breast tumors by increasing the expression of molecules with pro-tumorigenic activity such as interleukin-8 (IL-8) [55].

Until recently, it was believed that breast microcalcifications were only a degenerative process associated with tumorigenesis. Breast tumors have been shown to have both amorphous and crystalline deposits. Busing et al. suggested that amorphous deposits agree well with the interpretation that they are of a secondary dystrophic or degenerative nature [21,54].

After various studies, it is possible to “see” microcalcifications as an active biological process similar to the physiological mineralization occurring in bone [16].

In the model described by Morgan et al., HA crystals are formed in the cytoplasm of breast cancer cells and then released into extracellular space where they will form the mammary microcalcifications [22,54]. Moreover, in a recent in vitro study, O’Grady S. et al. described the role of both alkaline phosphatase (ALP) activity and dysregulation of the transient receptor potential cation channel, subfamily M (TRPM7) ion channel in the microcalcifications production [20,55]. These data support the evidence of the active role of breast cancer cells in the formation of microcalcifications also demonstrating the involvement of TRPM7 mediated Ca2+ transport.

In the context of breast cancer, it has been discovered that microcalcifications are actively produced by cells which, subjected to the EMT process, acquire a mesenchymal phenotype; these cells, via BMP-2 stimulation, can acquire an osteoblastic-like phenotype, becoming BOLCs. BOLCs displayed the capability to synthesize and secrete microcalcifications made of HA. For the first time, our group reported the presence of Mg-HAp, which was frequently observed in breast cancer but never found in benign lesions.

BOLCs express typical biomarkers of bone cells—such as BMP-2, BMP-4, osteopontin (OPN), RANKL, VDR, RUNX2, and PTX3. The expression of biomarkers such as RANKL and RUNX2 appears to be linked to the metastatic process of breast to bone. Our group also showed that BOLCs (RUNX2- RANKL double positive breast cancer cells) significantly increase in estrogen receptor (ER) positive breast cancers rather than human epidermal growth factor receptor-2 (HER2) positive ones. [54]. This strengthens the idea of the possible link between the origin of BOLCs and bone metabolism. Indeed, it is well known that estrogens are key mediators of osteoblast differentiation and function [56]. Therefore, ER overexpression in breast cancer could participate in the differentiation of tumor cells into osteoblast-like cells [55].

Besides, a recent study displayed a correlation between the presence of BOLCs and the development of bone metastatic lesions within five years from histological diagnosis [28,57]. Li Zhang et al. discovered that the tumor tissue produces BMP-2 locally through autocrine or paracrine ways, which combines with related receptors and enters the nucleus, acts at the target gene, and then regulates RUNX2, OPN, and other bone matrix proteins to form a microenvironment of calcium and phosphorus deposits in the tumor tissue, resulting in the formation of microcalcifications [51].

It is important to note that the here described mechanisms are not able to explain the production of all forms of breast microcalcifications. Indeed, clusters of microcalcifications are also found in patients without associated nodules where they represent the first sign of malignant tumor in the initial phase which over time becomes invasive [58]. In addition, it is observed that not all aggressive tumors, such as triple negative breast cancers, develop microcalcifications [59]. The osteoblastic differentiation of cancer cells can be characteristic of specific subtypes of breast cancer such as lumina A [26]. Therefore, the identification of biomarkers for early prediction of these types of breast lesions could improve the management of breast cancer patients. Lastly, algorithms of artificial intelligence capable to assess the risk of the presence of malignant lesions associated to the imaging characteristics of breast microcalcifications have been developed. In this context, Fanizzi et al. [60] proposed a highly performed automatic tool for characterizing and discriminating clusters of microcalcifications into benign/malignant in digital mammograms. Figure 2 shows histological and ultrastructural characteristics of breast microcalcifications.

## 3. Microcalcifications in Thyroid Lesions

Thyroid calcification has a very frequent occurrence involving about 40% of malignant lesions and 20% of benignant ones [61]. Some studies have reported that the features of both categories of lesions partially overlap [62] so it could be particularly challenging to identify malignant nodules as such. Also, many diseases—such as papillary carcinoma, multinodular goiter, sarcoidosis, and Graves’ disease—lead to various patterns of calcification. Generally, thyroid nodules are checked with ultrasonography (US) and this is usually the method involved in occasional findings [63]. Calcium deposits can be present in different shapes such as microcalcifications (echogenic foci of 1 mm or less); macrocalcification (bigger lesions up to an entire calcified nodule); rim calcification (eggshell calcification at the nodule margin) [64]. In addition, Peng et al. recently reported a special conformation of calcification defined ‘petal-like’, which seems to only occur in malignant lesions with a high incidence of lymph node metastasis. This pattern of calcification appears as numerous scattered hyperechogenic spots around a solid thyroid nodule with a cystic-like dark area ahead of each spot, defined as “black and white foci” [65]. Aside from these categories, there are isolated or undefined calcifications and these shouldn’t be considered benign because they can show a low to intermediate malignancy risk [66].

Overall, different guidelines consider hypo-echogenicity of lesions, irregular or lobulated margins, intranodular vascularity, and the presence of microcalcification as features of a high-risk nodule [67,68]. The hypothesis that the different compositions can affect the nature of the lesion and have possible implications on a differential diagnosis, cuts a path to more detailed studies. For example, Guerlain et al. [69] characterized calcifications in various thyroid diseases and identified their mineral composition according to their localization. The results report that colloid calcifications are characterized by CO deposition, while calcium phosphate is prevalent in capsular calcification and in PBs. Besides, calcium phosphate is associated with malignant tumors. The same team claims to indirectly identify a crystal clearance mechanism, considering the presence of CD44 and OPN but this affirmation surely needs further insights. It is noteworthy that calcification in PBs is represented by concentric structures suggesting a slow mineral deposition. PBs are largely considered the results of a dystrophic calcification occurring in dying tissues. A single PB is formed when a single necrotic cell develops lamellated calcifications, but they might represent an active biologic process leading to the degeneration of tumor cells. PBs are observed most commonly in papillary thyroid carcinoma (PTC), meningioma, and papillary serous cystadenocarcinoma of the ovary [70]. De Santis et al. closely investigated PBs microcalcification in thyroid cancer tissue. The group used several different analytical techniques—including energy dispersive X-ray analysis (EDX), scanning electron microscopy (SEM), and micro-Fourier transform infrared (micro-FTIR)—and summarized the results of these applied methods, found out that the mineral arrangement within the PBs is a composition of Ca, P, Mg, Na, Fe, and Zn [71]. Calcification is also important for the diagnosis of PTC and this is often found in PBs. PTC patients with microcalcifications also express higher levels of RUNX2 mRNA in serum than those without microcalcifications [72]. RUNX2 is a principal transcription factor associated with osteogenic differentiation and it is significantly related to PTC calcification and invasiveness. Moreover, high RUNX2 expression has already shown to be predictive of breast cancer recurrence [73] and it has been reported for thyrocytes to have an increased expression of RUNX2 in PTC tissues [74]. Jin et al. studied the regulatory role of RUNX2 in PTC and screened several molecules to identify those affecting the most on its activity. The results state that RUNX2 is enhanced by Homeobox A9 (HOXA9). In fact, HOXA9 is an upstream regulator of RUNX2 and acts as a positive regulator of RUNX2 [75]. HOX family encode a class of numerous transcription factors, often dysregulated in tumors [76] and exert their major role in cell growth and differentiation, regulating numerous genes. Interestingly, RUNX2-knockdown cells with HOXA9 overexpression exhibit enhanced migration and invasion compared to those in RUNX2-knockdown control cells. These findings support that HOXA9 is linked to the calcification also in an independent way from RUNX2. According to these molecular data, it is known that PTC often tends to metastasize to local cervical lymph nodes; lymph-node metastases and recurrence rates are approximately 50% and 20%, respectively [77]. These metastatic cells sometimes promote a calcification process at this site [78,79].

## 4. Microcalcifications in Lung Pathogenesis

Pulmonary calcification is an event that occurs with several systemic and pulmonary conditions, and it may be a marker of disease severity and its chronicity [80]. Pathologic soft tissue calcification can be broadly divided into metastatic calcification, in which calcium deposits in normal tissues, which are further subdivided into benign causes like chronic renal insufficiency on hemodialysis or hypervitaminosis D, and malignant causes like parathyroid carcinoma or multiple myeloma; and dystrophic calcification, in which calcification is superimposed on previously injured lung. The mechanisms that lead to the calcification process in the lung are not fully known, and there is no single factor responsible for this phenomenon. Ectopic metastatic calcification can be influenced by serum calcium and phosphate concentration, ALP activity, and local physicochemical conditions such as pH [80]. On the other hand, dystrophic calcification, by definition, requires injured tissues for calcification to occur, it is a local, organized process with local deposition of crystalline hydroxyapatite calcium salt where the serum level of calcium and phosphate is normal [81].

The etiology of the calcifications can be found by computed tomography (CT) or chest radiography which give information on the character of the calcifications. A central nidus is a sign of a granulomatous lesion, popcorn-ball calcification is characteristic of hamartoma and multiple punctate lesions throughout a lesion may be seen in either granulomas or hamartoma. Usually, calcifications in a pulmonary nodule are indicative of a benign process, but there are some exceptions: carcinoma may grow from an old calcified granuloma, or it may be a natural process of metastatic osteogenic sarcoma and chondrosarcomas [82,83].

Regarding metastatic calcifications, they are caused by high levels of serum calcium and phosphate which deposit in normal lung tissue. Common etiologies are primary or secondary hyperparathyroidism, chronic renal failure, and neoplasia leading to destructive bone lesions such as multiple myeloma. In the lungs, calcifications usually lodge in the interstitium of the alveolar septum, in the walls of the bronchioles and pulmonary vessels, or in the large airways. The upper lobes are mainly involved because of the higher ventilation ± perfusion ratio producing a lower end-capillary PCO2 and a higher pH, while the relatively base lobes have a higher end-capillary PCO2 and then a lower pH. The clinical manifestation of pulmonary calcifications is minimal but, it can happen that it is dyspnea [83]. In this context, Inomata et al., reported a case of miliary brain metastases with calcification of lung cancer. It is a rare metastatic form showing a miliary spread of punctate tumor nodules in the brain. Cancer cells are distributed in the perivascular spaces, without edema around them and without forming masses [84,85]

## 5. Microcalcifications in Ovary Cancer

The presence of calcification in ovaries is identified from radiological reports and occurs in a wide range of both primary neoplasms and metastatic diseases but it is generally associated with a lower histologic grade and indicates a poorer survival rate [85,86]. Ovarian cancer is the fifth leading cause of cancer-related death among women in the United States [87]. Calcification in ovarian cancer is mainly psammomatous and occurs in 8% ovarian cancer, prevalently in serous papillary ovarian cancer, whereas it is infrequent in ovarian adenocarcinoma [85]. As observed in PTC, PBs have a concentric shape formed by collagen production of cancer cells and subsequent calcification. Calcium deposits can be detected also in non-neoplastic conditions such as infarction, or mature teratomas [88] but the mechanism by which calcification forms is still poorly explored. The questions emerge around the significance of calcification in ovarian cancer’s diagnosis or differentiated therapy. It is reasonable to affirm that calcification arises where a local calcium/phosphorus metabolism is imbalanced. The entire process resembles an ectopic calcification, but it seems unclear if it is simply derived from a cellular degeneration or it is the result of part of an active process of calcification [70]. In this regard, Wen et al. suggest that ovarian cancer cells switch to a calcifying phenotype and express bone related transcription factors and bone matrix proteins to cause the calcification deposition [89]. Thus, the ovarian cancer cells are capable of switching their phenotype to osteoblast-like phenotype, experiencing a similar process that breast cancer cells undergo. Acquiring the osteoblast-like phenotype means acquiring the high affinity for the bone environment, which may finally explain the high frequency of bone metastasis derived from breast cancer [90]. In this case, the team used an ovarian cancer calcification model in vitro based on human ovarian cancer cells (SKOV3) and revealed the presence of cells resembling osteoblasts characteristics. In order to supervise the calcifying phenotype of SKOV3 cells during the experimental steps, they evaluated the expression of the mineralization biomarkers Collagen I, RUNX2, β2-microglobulin, BMP-2, Osterix, and OPN and there was a statistically significant difference of the expression of BMP-2 and OPN between the calcified group and non-calcified group [89]. Overall, calcification in ovarian cancer has a significance in clinical practice, but more attention should be paid to the mechanism of calcification. Given the influence that female reproductive apparatus endures, Das and colleagues [70] already affirmed that the formation of calcifications may be linked to the secretion of collagen, but it could be also induced by one or multiple hormones. In experimental models, they observed that calcifications can be induced through hormone administration, especially prolactin [91], but there are no recent updates in this respect.

The main molecules involved in the formation of ectopic calcifications in breast, prostate, ovary and thyroid tissues are summarized in Table 1.

## 6. Conclusions

Recent studies highlighted some of the molecular mechanisms involved in the formation of ectopic calcifications demonstrating a surprising similarity with processes that occur in the bone during the deposition of HA crystals. These mechanisms were frequently associated with the progression of human pathologies such as cancers and atherosclerosis. Among these molecular mechanisms, the EMT phenomenon could represent the junction ring between the origin of cells capable of producing the ectopic calcifications and the presence of them in non-bone tissues. Thus, the presence of calcifications in human tissues can be considered an early predictor for the occurrence or development of a malignant condition. Indeed, it is known that the EMT phenomenon is associated with the origin of human pathologies, as well as the formation of metastatic lesions.

These evidences could drastically change the current point of view concerning the role of ectopic calcifications in the progression of human diseases and also in the patients’ management. In fact, the presence of calcifications can suggest an unfavorable prognosis due to dysregulation of normal tissues homeostasis.

In conclusion, the discovery of the cellular and molecular signaling related to the formation of ectopic calcifications can lay the foundation for identifying new targets for developing innovative diagnostic and therapeutic approaches based on the presence/absence of these calcium deposits.

## Figures and Tables

**Figure 1 ijms-22-00120-f001:**
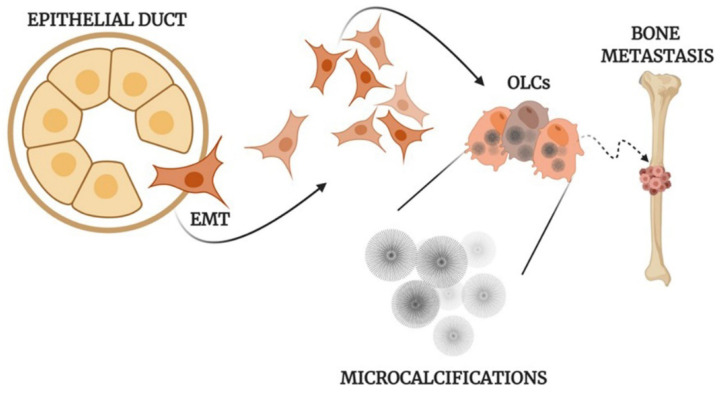
Mechanisms of the formation of calcifications. Schematic representation of osteoblast differentiation, microcalcification production, and metastasis development in human epithelial tumors. EMT: Epithelial to mesenchymal transition; OLCs: Osteoblast-Like Cells created with BioRennder.com.

**Figure 2 ijms-22-00120-f002:**
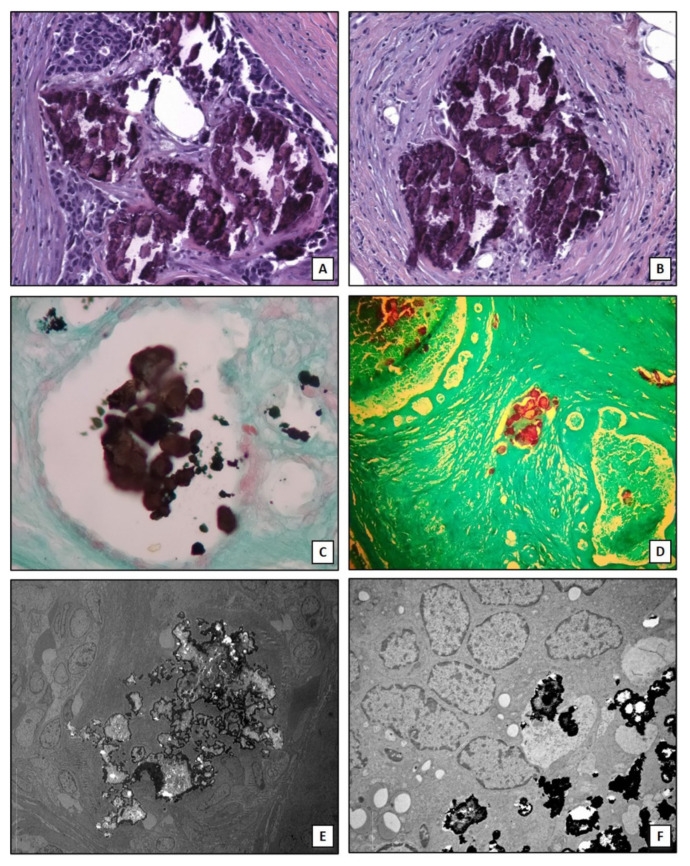
Breast microcalcifications. (**A**) Hematoxylin and eosin section shows dystrophic calcifications embedded into an infiltrating breast carcinoma (10×). (**B**) Dystrophic calcifications in breast a benign lesion (10×). (**C**) Traditional von Kossa staining shows calcium crystals (black) into a breast normal duct (10×). (**D**) TorVergata modified von Kossa staining displays calcium deposits (red) into breast ducts and stroma (5×). (**E**) Electron micrograph shows electrondense large calcification in an infiltrating breast carcinoma (2000×). (**F**) Small calcium crystals in an infiltrating breast carcinoma (5000×).

**Table 1 ijms-22-00120-t001:** Main reported biomarkers of calcification process in breast, thyroid, lung and ovarian tissues.

Biomarkers	Organs	Role	Localization	Refs.
BMP-2	Breast, prostate, ovary	Induces cartilage and bone formation	Vesicles, extracellular	[26,52,55,89]
HOXA9	Thyroid	Cell growth and differentiation	Nucleus	[75,79]
OPN	Breast, ovary	Provides the basic structure of the bone matrix	Golgi apparatus, extracellular	[52,55,89]
PTX3	Breast, prostate	Osteoblast proliferation, differentiation and function	Membrane, extracellular	[26,34,55]
RANKL	Breast, prostate	Osteoclatogenesis	Membrane, extracellular	[26]
RUNX2	Breast, prostate, thyroid	Osteoblast differentiation	Nucleus, Nucleoplasm	[26,52,55,72,73]
VDR	Breast, prostate	Modulation of calcium levels in the body	Nucleus	[26,34,55]

BMP2: Bone Morphogenetic Proteins-2; HOXA9: Homeobox protein Hox-A9; OPN: Osteopontin; PTX3: Pentraxin 3; RANKL: Receptor activator of nuclear factor kappa-Β ligand; RUNX2: Runt-related transcription factor 2; VDR: Vitamin D Receptor.

## Data Availability

Not applicable.

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
