# Peer review of "Molecular Aspects and Prognostic Significance of Microcalcifications in Human Pathology: A Narrative Review"

_ijms, 2020, doi:10.3390/ijms22010120_

Round 1
Reviewer 1 Report
Please make the corrections listed in the comments of my pdf file, attached.

Author Response
Ref.: Manuscript: ijms-1039463
"Molecular Aspects and prognostic significance of microcalcifications in human pathology: a narrative review "
Submitted to: International Journal of Molecular Sciences – SI: Calcification in Human Pathology
Before we begin the point-by-point review of the list of concerns, we would like to thank the Reviewer for their comments on how to improve the manuscript, which has been revised accordingly, as well as the Editors for calling for a new submission of an improved version of our manuscript.
Reply to Reviewer 1
Please make the corrections listed in the comments of my pdf file, attached.
Reply: we would like to thank the Reviewer for expressing interest in our work, and for their availability to review a revised version of our manuscript. In the revised form of our manuscript, we included all changes suggested by the reviewer.
Reviewer 2 Report
The work summarizes the characteristics and pathogenetic mechanisms of vascular calcifications and in some types of tumors, highlighting their potential as prognostic biomarkers.
The introduction clearly delineates the problem, the description of the mechanisms is detailed, the literature well reported.
Below I ask some questions and suggest some changes that could further improve the manuscript:
1) Lines 45-47 “These investigations reported possible molecular mechanisms involved in the deposition of calcium crystals, mainly the epithelial to mesenchymal transition (EMT) phenomenon and osteoblastic differentiation”
The mechanism explained is very interesting and in line with what some works of the literature affirm. On breast tumors, however, some clusters of microcalcifications without associated nodules represent the first sign of malignant tumor in the initial phase (in situ) which over time becomes invasive. However, it is observed that not all tumors develop microcalcifications and those that show less are some of the most aggressive tumors (triple negative, G3): how do the authors explain this aspect? Is it possible that the faster transition from healthy tissue to invasive carcinoma of the latter skips the in situ phase? Or that the absence of microcalcifications concerns some specific characteristics of these tumors? If so, which ones?
2) Lines 159-160 “However, currently mammographic devices are not able to discriminate different form of calcifications (e.g. type I and II microcalcifications)”
However, there are different radiological classifications that describe microcalcifications in the breast, classify them and offer a predictive evaluation of malignancy through the BIRADS system:
3) Although histological examination remains the gold standard for defining the nature of microcalcifications in breast carcinoma, it can be predicted not only by the human reader but today also by automatic systems trained to aid radiological reporting called CAD that intervene both in the detection and in the differentiation between benign and malignant microcalcifications. In this regard, I recommend that you integrate your references with this and other references:
-
Fanizzi A. et al. “Ensemble DiscreteWavelet Transform and Gray-Level Co-Occurrence Matrix for Microcalcification Cluster Classification in Digital Mammography”.Appl. Sci. 2019, 9, 5388; doi:10.3390/app9245388
4) even if there are some hints on the lung and ovary, in the article there is little mention of microcalcifications in metastases, especially in atypical sites. On this topic I suggest you add references to reflect this.
5) The differentiation process is described in great detail but in my opinion it would be appropriate to represent it also in succession with a visual scheme
Author Response
Ref.: Manuscript: ijms-1039463
"Molecular Aspects and prognostic significance of microcalcifications in human pathology: a narrative review "
Submitted to: International Journal of Molecular Sciences – SI: Calcification in Human Pathology
Before we begin the point-by-point review of the list of concerns, we would like to thank the Reviewer for their comments on how to improve the manuscript, which has been revised accordingly, as well as the Editors for calling for a new submission of an improved version of our manuscript.
Reply to Reviewer 2
The work summarizes the characteristics and pathogenetic mechanisms of vascular calcifications and in some types of tumors, highlighting their potential as prognostic biomarkers.
The introduction clearly delineates the problem, the description of the mechanisms is detailed, the literature well reported.
Reply: we would like to thank the Reviewer for expressing interest in our work, and for their availability to review a revised version of our manuscript.
1) Lines 45-47 “These investigations reported possible molecular mechanisms involved in the deposition of calcium crystals, mainly the epithelial to mesenchymal transition (EMT) phenomenon and osteoblastic differentiation”
The mechanism explained is very interesting and in line with what some works of the literature affirm. On breast tumors, however, some clusters of microcalcifications without associated nodules represent the first sign of malignant tumor in the initial phase (in situ) which over time becomes invasive. However, it is observed that not all tumors develop microcalcifications and those that show less are some of the most aggressive tumors (triple negative, G3): how do the authors explain this aspect? Is it possible that the faster transition from healthy tissue to invasive carcinoma of the latter skips the in situ phase? Or that the absence of microcalcifications concerns some specific characteristics of these tumors? If so, which ones?
Reply: Thank you for this pointing out. In the new version of our manuscript, we discussed these apparent discordances in the paragraph “2. Microcalcifications in breast cancer”
Pag. 5 and 6
It is important to note that the here described mechanisms are not able to explain the production of all forms of breast microcalcifications. Indeed, clusters of microcalcifications are also found in patients without associated nodules where they represent the first sign of malignant tumor in the initial phase which over time becomes invasive [58]. In addition, it is observed that not all aggressive tumors, such as triple negative breast cancers, develop microcalcifications [59]. The osteoblastic differentiation of cancer cells can be characteristic of specific subtypes of breast cancer such as lumina A [26]. Therefore, the identification of biomarkers capable for early prediction of these types of breast lesions could improve the management of breast cancer patients. Lastly, algorithms of artificial intelligence capable to assess the risk of the presence of malignant lesions associated to the imaging characteristics of breast microcalcifications have been developed. In this context, Fanizzi et al. [60] proposed an highly performed automatic tool for characterizing and discriminating clusters of microcalcifications into benign/malignant in digital mammograms. Figure 2 shows histological and ultrastructural characteristics of breast microcalcifications.
2) Lines 159-160 “However, currently mammographic devices are not able to discriminate different form of calcifications (e.g. type I and II microcalcifications)”
However, there are different radiological classifications that describe microcalcifications in the breast, classify them and offer a predictive evaluation of malignancy through the BIRADS system:
Reply: Thank you for this pointing out. In the paragraph “2. Microcalcifications in breast cancer” we discussed it.
Pag.5
Microcalcifications play a crucial role in early breast cancer diagnosis. In fact, approximately 50% of non-palpable breast cancers are detected by mammography only through microcalcifications patterns [27,51], but mammography cannot distinguish type I and II microcalcifications. However, some classification systems, such as Breast Imaging Report and Data System (BIRADS) [52], associate the shape, density, and distribution within the breast of the microcalcifications to assess the risk of the presence of a malignant lesion. Nevertheless, a tissue biopsy should be performed to confirm if the microcalcifications seen on mammography are associated with cancer. Analysis of microcalcifications by electron microscopy, microprobe analysis, and X-ray diffraction, has revealed two distinct forms of microcalcifications in breast disease based on their appearance and chemical composition [21].
3) Although histological examination remains the gold standard for defining the nature of microcalcifications in breast carcinoma, it can be predicted not only by the human reader but today also by automatic systems trained to aid radiological reporting called CAD that intervene both in the detection and in the differentiation between benign and malignant microcalcifications. In this regard, I recommend that you integrate your references with this and other references:
Fanizzi A. et al. “Ensemble DiscreteWavelet Transform and Gray-Level Co-Occurrence Matrix for Microcalcification Cluster Classification in Digital Mammography”.Appl. Sci. 2019, 9, 5388; doi:10.3390/app9245388
Reply: we discussed this in the new version of our manuscript.
4) even if there are some hints on the lung and ovary, in the article there is little mention of microcalcifications in metastases, especially in atypical sites. On this topic I suggest you add references to reflect this.
Reply: Thank you for this pointing out. This phenomenon is rarely reported in the literature. However, we added a little discussion concerning microcalcifications in metastases in both “3. Microcalcifications in thyroid lesions” and “4. Microcalcifications in lung pathogenesis” paragraphs.
5) The differentiation process is described in great detail but in my opinion it would be appropriate to represent it also in succession with a visual scheme
Reply: Thank you for this pointing out. We added a visual scheme concerning the differentiation of epithelial cells in osteoblast like cells and the formation of microcalcifications.
Round 2
Reviewer 2 Report
Thanks for responding to my suggestions. In my opinion, the work is now ready for publication. Congratulations